# Towards Accurate and Calibrated Classification: Regularizing Cross-Entropy From A Generative Perspective

## Abstract

Accurate classification requires not only high predictive accuracy but also well-calibrated confidence estimates. Yet, modern deep neural networks (DNNs) are often overconfident, primarily due to overfitting on the negative log-likelihood (NLL). While focal loss variants alleviate this issue, they typically reduce accuracy, revealing a persistent trade-off between calibration and predictive performance. Motivated by the complementary strengths of generative and discriminative classifiers, we propose Generative Cross-Entropy (GCE), which maximizes $p(x|y)$ and is equivalent to cross-entropy augmented with a class-level confidence regularizer. Under mild conditions, GCE is *strictly proper*. Across CIFAR-10/100, Tiny-ImageNet, and a medical imaging benchmark, GCE improves both accuracy and calibration over cross-entropy, especially in the long-tailed scenario. Combined with adaptive piecewise temperature scaling (ATS), GCE attains calibration competitive with focal-loss variants without sacrificing accuracy.

## 1 Introduction

Accurate and reliable predictive probabilities are essential for deploying machine learning models in high-stakes real-world applications, where confidence estimates directly affect downstream decision-making (e.g., medical diagnosis, autonomous driving). A well-calibrated model produces probability estimates that faithfully reflect the true likelihood of correctness: for instance, predictions made with 80% confidence should be correct approximately 80% of the time. However, despite their remarkable success in terms of accuracy, modern deep neural networks (DNNs) are notoriously miscalibrated, often exhibiting severe overconfidence. This phenomenon has been attributed to their tendency to overfit the negative log-likelihood (NLL) during training (Mukhoti et al., 2020), ultimately undermining the trustworthiness of their probabilistic predictions.

Recent research has sought to mitigate miscalibration by either introducing differentiable calibration regularizers or designing calibration-aware loss functions. For example, Maximum Mean Calibration Error (MMCE) minimizes a kernel-based discrepancy (Kumar et al., 2018), though it is sensitive to kernel choice and faces optimization challenges. The Brier Score (BS) (Hui & Belkin, 2020), a proper scoring rule, provides modest gains but generally underperforms specialized losses. Among loss-based methods, focal loss (FL) (Mukhoti et al., 2020) improves calibration by down-weighting easy examples to counteract overconfidence. More advanced variants, such as Dual Focal Loss (DFL) (Tao et al., 2023) and Adaptive Focal Loss (AFL) (Ghosh et al., 2022), further refine this balance through adjustments based on competing logits or adaptive focusing parameters. Nevertheless, a persistent drawback of these approaches is that calibration improvements often come at the expense of predictive accuracy, reflecting a fundamental trade-off between the two objectives.

This tension motivates a natural question: *can we improve calibration without sacrificing accuracy*? To address this, we revisit the longstanding contrast between discriminative and generative classifiers. While discriminative models typically excel in accuracy, generative models are more robust to overfitting (Li et al., 2019; Lee et al., 2019), which often leads to better-calibrated predictions. Building on this insight, we bring a generative perspective into discriminative training: maximize the log-likelihood of the posterior $p(x|y)$. We show that this objective is equivalent to augmenting cross-entropy with a class-level confidence constraint, thereby encouraging calibrated predictions

while preserving the discriminative nature of the classifier. To the best of our knowledge, this is the first work to explicitly leverage a generative perspective to design a *strictly proper* loss that improves both accuracy and calibration.

Beyond the proposed Generative Cross-Entropy (GCE) loss, we introduce an adaptive piecewise temperature scaling technique tailored to further improve post-hoc calibration. Through extensive experiments on CIFAR-10/100, Tiny-ImageNet, and a medical imaging benchmark, we demonstrate that our method consistently enhances both accuracy and calibration compared to standard cross-entropy. Moreover, when combined with post-hoc calibration, GCE achieves calibration performance competitive with state-of-the-art focal loss variants, while preserving—or even improving—accuracy. These results establish GCE as a principled and practical approach to achieving a more favorable accuracy–calibration trade-off.

## 2 RELATED WORK

Recent advances in addressing miscalibration in deep neural networks can be broadly divided into two paradigms: *training-time loss modification* and *post-hoc calibration*.

**Training-Time Loss Modification.** A first line of research introduces differentiable calibration-aware objectives directly into the training process. For instance, Maximum Mean Calibration Error (Kumar et al., 2018) minimizes a kernel-based measure of miscalibration, though its performance is highly sensitive to kernel choice and associated optimization challenges. The Brier Score (Hui & Belkin, 2020), a proper scoring rule, computes the mean squared error between predicted probabilities and one-hot targets, offering moderate calibration gains but typically underperforming more specialized losses. Label Smoothing (Müller et al., 2019) softens the target distribution, thereby mitigating overconfidence. More recently, a series of specialized loss functions have emerged: Focal Loss (Mukhoti et al., 2020) downweights easy examples to counteract overconfidence, Inverse Focal Loss penalizes underconfidence, Dual Focal Loss (Tao et al., 2023) balances both by incorporating the highest non-target logit, and Adaptive Focal Loss (Ghosh et al., 2022) dynamically tunes the focusing parameter $\gamma$ based on validation feedback. Collectively, these methods achieve strong calibration improvements but often at the cost of reduced accuracy, highlighting the inherent trade-off between calibration and discriminative performance.

**Post-Hoc Calibration.** A second paradigm performs lightweight transformations on the outputs of a pretrained model using held-out data. Classic methods include Platt Scaling (Platt et al., 1999), Isotonic Regression (Zadrozny & Elkan, 2002), and Histogram Binning (Zadrozny & Elkan, 2001), with extensions such as Bayesian Binning into Quantiles (BBQ). Among these, Temperature Scaling (Guo et al., 2017) has become the de facto standard due to its simplicity and strong empirical efficacy. More recent variants, such as Entropy-based Temperature Scaling and Parameterized Temperature Scaling, provide finer-grained adaptive adjustments. Post-hoc methods have the advantage of preserving predictive accuracy while substantially improving calibration, making them a widely adopted complement or alternative to training-time strategies.

## 3 PRELIMINARY

### 3.1 PROBLEM FORMULATION

Let $\mathcal{D} = \{(x_i, y_i)\}_{i=1}^N$ denotes a dataset consist of $N$ *i.i.d.* samples from the joint distribution $\mathcal{P}(\mathcal{X} \times \mathcal{Y})$, where for each sample $i$, $x_i \in \mathcal{X}$ is the input and $y_i \in \mathcal{Y} = \{1, 2, \cdots, K\}$ is the corresponding class label. In a classification task, we seek a function $f : \mathcal{X} \to \mathbb{R}^K$ that maps each input $x \in \mathcal{X}$ to a vector of class logits $z = f(x) \in \mathbb{R}^K$. These logits are then converted into a probability distribution $\hat{p} = \mathrm{softmax}(z) \in \Delta^K$ over the $K$ classes, where $\hat{p}_i = \frac{\exp(z_i)}{\sum_{j=1}^K \exp(z_j)}$. We refer to $\hat{y} = \arg\max_{i \in \mathcal{Y}} \hat{p}_i$ as the classifier's prediction and $\hat{p}_{\hat{y}}$ as its associated confidence.

**Loss Function** The classifier $f$ is usually learned by minimizing a certain loss function over a pre-defined hypothesis class. In the scenario of modern deep learning, the hypothesis class is a parametrized neural network $f_\theta$. Typically, a loss function $\ell : \Delta^K \times \Delta^K \to \mathbb{R}$ is defined point-wise,

where $\Delta^K$ denotes the $(k-1)$-dimensional probability simplex. For example, the cross-entropy and focal loss per sample is defined as $\ell_{\text{CE}}(\hat{p}^{(i)}, q^{(i)}) = \sum_{k=1}^{K} q_k^{(i)} \log \hat{p}_k^{(i)}$ and $\ell_{\text{FL}}(\hat{p}^{(i)}, q^{(i)}) = \sum_{k=1}^{K} q_k^{(i)} (1 - \hat{p}_k^{(i)})^\gamma \log \hat{p}_k^{(i)}$ respectively, where $q^{(i)}$ is the one-hot encoding of the ground truth label $y_i$. Denote $\hat{\mathbf{P}} = (\hat{p}^{(1)}, \cdots, \hat{p}^{(N)})$ and $\mathbf{Q} = (q^{(1)}, \cdots, q^{(N)})$. Then the total loss (empirical risk) $\mathcal{L} : (\Delta^K)^N \times (\Delta^K)^N \to \mathbb{R}$ is computed by averaging $\ell$ over all samples, i.e., $\mathcal{L}(\hat{\mathbf{P}}, \mathbf{Q}) = \frac{1}{N} \sum_{i=1}^{N} \ell(\hat{p}^{(i)}, q^{(i)})$.

**Classification Calibration**    A classifier is said to be perfectly calibrated if, for any predicted confidence score $p \in [0, 1]$, the conditional accuracy given $p$ is exactly $p$, i.e., $\mathbb{P}(\hat{y} = y | \hat{p} = p) = p$. Intuitively, among all samples for which the classifier assigns a confidence of 80%, exactly 80% should be correctly classified. The most commonly used metric to evaluate model calibration is the *expected calibration error* (ECE) (Naeini et al., 2015), defined as $\mathbb{E}_{\hat{p}}[|\mathbb{P}(\hat{y} = y | \hat{p}) - \hat{p}|]$. In practice, the predictions are partitioned into $M$ equally-spaced bins, and ECE is computed as the weighted average of the difference between accuracy and confidence in each bin. Formally,

$$\text{ECE} = \sum_{m=1}^{M} \frac{|B_m|}{N} \Big| \text{conf}(B_m) - \text{acc}(B_m) \Big|, \tag{1}$$

where $\text{conf}(B_m) = \frac{1}{|B_m|} \sum_{i \in B_m} \hat{p}_k^{(i)}$ is the average confidence of bin $B_m$ and $\text{acc}(B_m) = \frac{1}{|B_m|} \sum_{i \in B_m} \mathbb{I}(\hat{y}_i = y_i)$ is the accuracy in bin $B_m$. Besides ECE, some other variants are also proposed to measure calibration error from different perspectives, such as AdaECE (Nguyen & O'Connor, 2015), which evenly distributes samples into each $B_m$, and ClasswiseECE (Kull et al., 2019), which approximates the ECE over $K$ classes.

**Temperature Scaling**    Given a model's logits vector $z = (z_1, \cdots, z_K)$ for $K$ classes, temperature scaling applies a scalar $T > 0$ to adjust confidence: $\hat{p}_i = \frac{\exp(z_i/T)}{\sum_{j=1}^{K} \exp(z_j/T)}$. If $T > 1$, the distribution is softened (higher entropy, less confident). If $T < 1$, it becomes sharper (lower entropy, more confident). $T = 1$ yields the original probabilities.

# 4 METHODOLOGY

## 4.1 ACCURACY VS. CALIBRATION

While many variants of focal loss have demonstrated strong performance in improving model calibration, they often come at the cost of reduced classification accuracy. It is important to emphasize that among the multiple dimensions used to evaluate a classifier, accuracy should generally take precedence over calibration. This is because achieving perfect calibration can be trivially accomplished by sacrificing predictive performance. For instance, consider a binary classification task where the dataset can be partitioned into two subsets: subset A contains 40 positive and 10 negative examples, and subset B contains 20 positive and 30 negative examples. A model that uniformly outputs a probability of 0.6 across all samples would achieve an Expected Calibration Error (ECE) of zero, yet only attain an overall accuracy of 60%. In contrast, a model that predicts 0.8 for samples in subset A and 0.4 for those in subset B achieves both perfect calibration and a higher accuracy of 70%. This discrepancy stems from the fact that the former model, despite its optimal ECE, fails to provide accurate estimates of the true class-posterior probabilities. This highlights the necessity of employing strictly proper loss functions, which are formally defined as follows.

**Definition 1** (Strictly Proper Loss). We say that a loss $\mathcal{L} : (\Delta^K)^N \times (\Delta^K)^N \to \mathbb{R}$ is strictly proper if $\mathcal{L}(\mathbf{P}, \mathbf{Q})$ is minimized if and only if $\mathbf{P} = \mathbf{Q}$.

Charoenphakdee et al. (2021) proved that *Focal loss* is not a strictly proper loss in general, thus not appropriate to estimate the true class posterior probability.

## 4.2 CROSS-ENTROPY AS MAXIMUM LIKELIHOOD ESTIMATION

A fundamental justification for the use of cross-entropy as a training objective lies in its equivalence to maximum likelihood estimation (MLE). Given a dataset $\mathcal{D} = \{(x_i, y_i)\}_{i=1}^{N}$, and a model parame-

terized by $\theta$ that defines conditional probabilities $p_\theta(y|x)$, the MLE framework seeks parameters that maximize the joint likelihood $\Pi_{i=1}^N p_\theta(y_i|x_i)$. Taking the negative logarithm yields the negative log-likelihood (NLL) $-\sum_{i=1}^N \log p_\theta(y_i|x_i)$, which coincides exactly with the cross-entropy loss when labels are encoded as one-hot vectors[1]. Thus, minimizing cross-entropy is mathematically equivalent to performing MLE under a categorical distribution, providing both a probabilistic interpretation and a principled statistical foundation for its widespread use in classification tasks. Besides, it has been widely known that cross-entropy is strictly proper (Charoenphakdee et al., 2021).

### 4.3 GENERATIVE CROSS-ENTROPY

While effective for training classification models, Cross-Entropy's purely discriminative nature can also pose limitations: the formulation neglects the generative structure of the data. Prior works (Ng & Jordan, 2001; Zheng et al., 2023) suggest that generative classifiers often converge faster and exhibit improved robustness against overfitting compared to their discriminative counterparts, as they leverage additional information about the data distribution. Motivated by this insight, we reformulate the training objective in terms of the posterior likelihood of $x$, i.e., maximizing $p(x|y)$. Concretely, we employ Bayes' rule to express

$$p_\theta(x|y) = \frac{p_\theta(y|x)p(x)}{p_\theta(y)} = \frac{p_\theta(y|x)p(x)}{\int p_\theta(y|x)p(x)\mathrm{d}x}, \tag{2}$$

which allows us to retain the discriminative modeling of $p_\theta(y|x)$ while implicitly incorporating the generative perspective through the normalization over classes. In practice, since the true prior distribution $p(x)$ is unknown, we approximate it with the empirical distribution of the dataset $\mathcal{D}$, i.e., $\hat{p}(x) = \frac{1}{N}\sum_{i=1}^N \delta(x - x_i)$. Thus, for a training sample $(x_i, y_i)$,

$$p_\theta(x_i|y_i) = \frac{p_\theta(y_i|x_i)\hat{p}(x_i)}{\sum_{i=1}^N p_\theta(y_i|x_i)\hat{p}(x_i)} = \frac{p_\theta(y_i|x_i)}{\sum_{i=1}^N p_\theta(y_i|x_i)}. \tag{3}$$

We then define the joint negative log-likelihood of $p_\theta(x|y)$, termed *Generative Cross-Entropy* (GCE) as

$$\mathcal{L}_{\text{GCE}} = -\frac{1}{N}\sum_{i=1}^N \log(\frac{p_{y_i}^{(i)}}{\sum_{j=1}^N \hat{p}_{y_i}^{(j)}}). \tag{4}$$

This design[2] enables our method to preserve the computational advantages of discriminative training while aligning the optimization process with the generative principle of modeling the posterior of $x$. Intuitively, this hybrid perspective mitigates the overfitting tendency of standard cross-entropy and encourages representations that are more faithful to the underlying data distribution.

Furthermore, if we denote $N_k = \sum_{i=1}^N \mathbb{I}(y_i = k)$, then GCE can be further written as

$$\mathcal{L}_{\text{GCE}} = -\frac{1}{N}\sum_{i=1}^N \log \hat{p}_{y_i}^{(i)} + \frac{1}{N}\sum_{k=1}^K N_k \cdot \log(\sum_{j=1}^N \hat{p}_k^{(j)}) \tag{5}$$

The first term in Eq. equation 5 corresponds exactly to the standard cross-entropy loss $\mathcal{L}_{\text{CE}}$. For each class $k$, we define the aggregated confidence as $C_k = \sum_{j=1}^N \hat{p}_k^{(j)}$, using this notation, the second term in Eq. equation 5 can be interpreted as a weighted sum of the logarithm of the aggregated confidence across all classes, where the weight for each class is proportional to its sample count $N_k$. Intuitively, while the first term penalizes misclassification at the individual-sample level, the second term regularizes the class-level confidence, thereby discouraging overconfident predictions. One advantage of GCE is that, under a minor assumption on the dataset, it is strictly proper:

**Theorem 1.** *Suppose $\boldsymbol{Q} = (q^{(1)}, \cdots, q^{(N)})$ has full row rank. The generative cross-entropy loss is strictly proper.*

*Proof.* See Appendix A ☐

---

[1] Here we ignore the normalizing factor $\frac{1}{N}$ since it does not affect the optimization problem.

[2] Like other loss functions, $\mathcal{L}_{\text{GCE}}$ is computed using mini-batch instead of the entire dataset.

**Mildness of the Assumption** In practical implementation, where the target distribution $q^{(i)}$ is taken as the one-hot encoding of the ground-truth label $y_i$, the full-row-rank assumption is automatically satisfied provided that each class appears at least once in the training set. Indeed, each row of the label matrix contains a single 1 in a unique column; thus, for any non-trivial linear combination of the rows, selecting the row corresponding to a non-zero coefficient immediately yields a contradiction, forcing all coefficients to vanish. Hence, the label matrix is of full row rank.

Notably, the conventional cross-entropy (CE) loss is known to be strictly proper only under this full-row-rank condition when one-hot labels are used. In mini-batch training, it may happen that not all classes are represented in a given batch, i.e., the set of classes appearing in the mini-batch $K_{\mathrm{mini}}$ is a strict subset of the full class set $K$. However, both CE and GCE are formulated such that the loss only depends on the probabilities assigned to the classes that actually appear in the batch. As a consequence, the analysis can be restricted to the effective classification task defined on $K_{\mathrm{mini}}$, where the same row-full-rank assumption continues to hold. Therefore, GCE inherits the same broad applicability as CE: the assumption underlying its strict propriety remains valid both in full-dataset training and in stochastic mini-batch training.

## 4.4 Adaptive Temperature Scaling

While temperature scaling is widely used for post-hoc calibration due to its simplicity and effectiveness, it applies a single global temperature to all samples, which limits its capacity to correct confidence miscalibration that varies across the confidence spectrum. In practice, modern neural networks often exhibit non-uniform calibration error: low-confidence and high-confidence predictions may be miscalibrated in different directions.

To address this limitation, we propose a piecewise temperature scaling method that adaptively calibrates classifier outputs based on equal mass confidence binning. Given a trained classifier and a held-out validation set, we first compute the predicted confidence scores and partition the data into bins using quantile-based thresholds. Each bin is associated with a separate temperature parameter that modulates the logits for samples assigned to that bin. During calibration, we iteratively update each bin's temperature using a gradient-free rule inspired by the confidence–accuracy gap. Specifically, if the average confidence within a bin exceeds the bin's empirical accuracy, the temperature is increased; otherwise, it is decreased. The update step is adaptively scaled by the magnitude of the calibration gap and clamped to a small range to ensure stability. This procedure is repeated for a fixed number of rounds. At each iteration, the calibration loss (e.g., ECE) is evaluated, and the best-performing temperature vector is retained. After convergence, the learned per-bin temperatures are used to rescale the model's logits at test time, improving calibration without degrading accuracy. The complete procedure is given in Algorithm 1.

---

**Algorithm 1:** ADAPTIVE TEMPERATURE SCALING

**Input:** Validation set $\mathcal{D}_{\mathrm{val}}$, number of bins $M$
**Output:** Updated per-bin temperatures $T_1, \ldots, T_M$
**Initialize:** Collect validation logits $\mathbf{z}$ and labels $y$; compute confidences $c_i = \max_j \mathrm{softmax}(z_i)_j$;
  compute per-bin thresholds $0 = \tau_0 < \cdots < \tau_M = 1$ as quantiles of $\{c_i\}$; assign bins index
  $b_i \leftarrow \mathrm{bucketize}(c_i; \{\tau_m\})$; initialize temperature vector $T = [1, \ldots, 1] \in \mathbb{R}^M$

**Training: for** $r \leftarrow 1$ **to** $R$ **do**
  **for** $m \leftarrow M - 1$ **to** $0$ **do**
    $\mathcal{I}_m = \{i : b_i = m\}$
    **if** $\mathcal{I}_m = \emptyset$ **then**
      **continue**
    $\tilde{z}_i \leftarrow z_i/T_m$; $c_i = \max_j \mathrm{softmax}(\tilde{z}_i)_j$
    $a_m = \frac{1}{|\mathcal{I}_m|} \sum_{i \in \mathcal{I}_m} \mathbb{I}[\arg\max_j \tilde{z}_{ij} = y_i]$; $c_m = \frac{1}{|\mathcal{I}_m|} \sum_{i \in \mathcal{I}_m} c_i$
    $\Delta \leftarrow \mathrm{clip}(\alpha(c_m - a_m), -0.1, 0.1)$; $T_m \leftarrow \mathrm{clip}(T_m + \Delta, T_{\min}, T_{\max})$
  $\tilde{z}_i \leftarrow z_i/T_{b_i}$; compute calibration error $\mathcal{L}_{\mathrm{ECE}}(\tilde{z}, y)$
  **if** *metric is best so far* **then**
    Save $T^* \leftarrow T$

**Return** $T^*$

---

## 5 EXPERIMENTS

We evaluate the performance of our method on datasets including CIFAR-10/100 (Krizhevsky et al., 2009), TinyImageNet (Deng et al., 2009), and Tau PET (AV1451), an imaging modality from the ADNI dataset measuring tau protein accumulation (Jagust et al., 2015). Models are trained using multiple deep neural network architectures, including ResNet-50, ResNet-110 (He et al., 2016), Wide-ResNet-28-10 (Zagoruyko & Komodakis, 2016), DenseNet121 (Huang et al., 2017), and a multi-layer perceptron. Further details on the dataset and implementation are given in Appendix B.

**Baselines.** We adapt multiple baseline methods, including cross entropy (CE), Brier loss (BS) (Hui & Belkin, 2020), MMCE (Kumar et al., 2018), focal loss (FLSD) (Mukhoti et al., 2020), dual focal loss (DFL) (Tao et al., 2023), and adaptive focal loss (AFL) (Ghosh et al., 2022).

**Training Setup.** We follow the same training settings as (Mukhoti et al., 2020), our method and baseline models are implemented based on the public code provided by Mukhoti et al. (2020). For CIFAR-10/100, models are trained for 350 epochs. From the training set, 5,000 images are reserved for validation. The learning rate schedule is as follows: 0.1 for the first 150 epochs, 0.01 for the subsequent 100 epochs, and 0.001 for the remaining epochs. For Tiny-ImageNet, training proceeds for 100 epochs, with a learning rate of 0.1 for the first 40 epochs, 0.01 for the following 20 epochs, and 0.001 thereafter. We conduct all experiments on a single NVIDIA A100 GPU and repeat each experiment 10 times with fixed random seeds 1-10 for reproducibility. Stochastic Gradient Descent (SGD) is employed with a momentum of 0.9 and a weight decay of $5 \times 10^{-4}$. For all datasets, both training and testing batch sizes are set to 128. For each baseline, we adapt the hyperparameter settings reported in the original paper to achieve the best performance on the corresponding dataset.

**Adaptive Temperature Scaling.** We use the same settings for our proposed adaptive temperature scaling across all experiments conducted in this work. Specifically, the confidence is partitioned into $n_{\text{bin}} = 15$ bins with initialized temperature $T_{\text{init}} = 1$. The update step coefficient $\alpha$ is set to 0.05. All temperatures are restricted in $[0.1, 10]$. The maximum number of rounds $R$ is set to 200.

**Evaluations.** To compare model accuracy, we report classification error rates (%). For calibration evaluation, we consider three complementary metrics: Expected Calibration Error (ECE), Adaptive ECE (AdaECE), and Classwise ECE. Each experiment is repeated 10 times with different random seeds for train/validation splits to ensure robustness and fairness of the results.

### 5.1 CLASSIFICATION ACCURACY

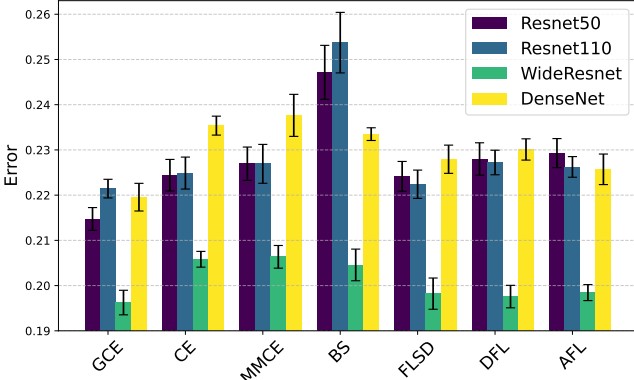

Figure 1: Classification error with 95% confidence intervals across four architectures on CIFAR-100. GCE consistently achieves lower error than CE and other baselines, confirming its effectiveness.

Table 1 reports the average test error across all datasets and architectures. Overall, GCE outperforms competing methods in the vast majority of settings, achieving state-of-the-art results in 8 out of 11 experiments. This trend is particularly evident on CIFAR-100, where, as illustrated in Figure 1, GCE achieves lower errors than all baselines across every evaluated architecture. The paired seed-wise one-sided tests (see Appendix C) further confirm that these improvements are statistically significant, highlighting the robustness of the method. Beyond standard vision benchmarks,

GCE also delivers strong performance on the AV1451 dataset, where it achieves the lowest test error among all compared methods. This demonstrates the broad applicability of our approach across different domains and data characteristics, not limited to canonical image classification tasks. We note, however, that on more challenging datasets such as Tiny-ImageNet, GCE does not surpass the three focal loss variants. This outcome is consistent with the original motivation of focal loss, which is explicitly designed to handle hard-to-classify examples. Importantly, despite a few exceptions, GCE outperforms standard cross-entropy (CE) in 9 out of 11 settings, establishing clear advantages in both moderate and complex tasks.

Table 1: Test error (%) over 10 runs. The lowest error is marked in bold.

| Dataset | Model | CE | Brier Loss | MMCE | FLSD | DFL | AFL | GCE |
|---|---|---|---|---|---|---|---|---|
| CIFAR-10 | ResNet-50 | 4.82±0.15 | 5.23±0.21 | 5.05±0.15 | 5.29±0.21 | 5.45±0.21 | 5.27±0.17 | **4.75±0.13** |
| | ResNet-110 | 5.12±0.26 | 5.10±0.15 | 5.20±0.18 | 5.11±0.19 | 5.40±0.15 | 5.24±0.15 | **4.98±0.17** |
| | Wide-ResNet | **4.06±0.10** | 4.37±0.11 | 4.19±0.12 | 4.20±0.18 | 4.08±0.09 | 4.48±0.15 | 4.10±0.13 |
| | DenseNet-121 | 5.11±0.17 | 4.94±0.11 | 5.35±0.16 | 5.19±0.17 | 5.51±0.31 | 5.25±0.09 | **4.93±0.10** |
| CIFAR-100 | ResNet-50 | 22.44±0.37 | 24.72±0.66 | 22.70±0.40 | 22.42±0.35 | 22.80±0.39 | 22.93±0.35 | **21.48±0.28** |
| | ResNet-110 | 22.48±0.39 | 25.37±0.72 | 22.69±0.47 | 22.24±0.33 | 22.71±0.29 | 22.62±0.25 | **22.15±0.21** |
| | Wide-ResNet | 20.58±0.18 | 20.45±0.38 | 20.63±0.28 | 19.82±0.37 | 19.76±0.27 | 19.85±0.19 | **19.63±0.30** |
| | DenseNet-121 | 23.54±0.22 | 23.35±0.17 | 23.76±0.50 | 22.78±0.34 | 23.01±0.25 | 22.56±0.38 | **21.96±0.33** |
| Tiny-ImageNet | ResNet-50 | 50.85±0.44 | 53.88±0.81 | 51.96±0.57 | 49.35±0.46 | 49.07±0.36 | **48.66±0.50** | 50.84±0.33 |
| | ResNet-110 | 48.14±0.46 | 50.40±0.93 | 49.93±0.80 | 46.72±0.32 | 47.03±0.38 | **46.59±0.33** | 49.44±0.41 |
| AV1451 | MLP | 36.78±0.51 | 36.51±0.67 | 37.56±0.85 | 37.89±0.37 | 38.67±0.37 | 37.78±0.76 | **36.44±0.87** |

## 5.2 CALIBRATION PERFORMANCE

We report the average Expected Calibration Error (ECE), computed with 20 bins, before and after applying adaptive temperature scaling in Table 2. Result for AdaECE and ClasswiseECE can be found in Appendix C. Two main observations emerge. First, GCE achieves superior calibration to standard cross-entropy (CE) both before and after temperature scaling, demonstrating its intrinsic advantage in reducing miscalibration. Second, although the raw calibration of GCE is not on par with state-of-the-art focal loss variants, applying appropriate post-hoc calibration allows GCE to reach comparable performance. Therefore, with a strong post-hoc calibrator, GCE matches focal-loss calibration while preserving accuracy, yielding a consistently better accuracy–calibration trade-off. To verify the effectiveness of our proposed adaptive temperature scaling (ATS) in improving calibration, we also compare it with other post hoc calibration methods (ensemble temperature scaling and spline fitting). Figure 2 visualizes the results for ResNet-50 on CIFAR-10. It shows that our proposed adaptive temperature scaling is a more effective post-hoc calibration method.

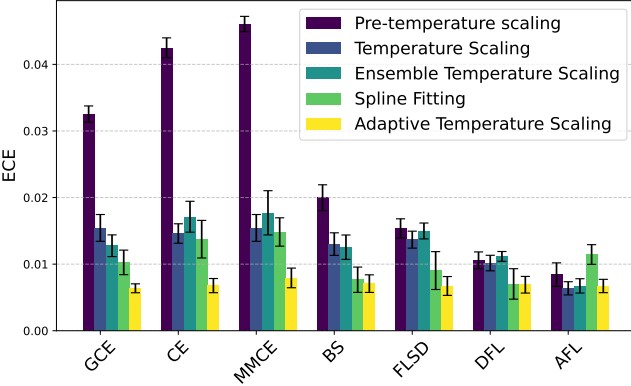

Figure 2: Expected Calibration Error (ECE) for ResNet-50 on CIFAR-10. Results are shown before scaling, with naive temperature scaling, and with Adaptive Temperature Scaling (ATS). GCE significantly reduces miscalibration relative to CE and reaches performance comparable to focal-loss variants with ATS.

We also notice that, on the more challenging Tiny-ImageNet dataset, GCE underperforms focal-loss variants in both classification accuracy and calibration. We attribute this to the specific design motivation of focal losses, which explicitly emphasize hard-to-classify examples by down-weighting easy samples. Tiny-ImageNet, with its large number of classes and visually similar categories,

Table 2: Average ECE (%) over 10 runs computed for different losses with adaptive temperature scaling (crossvalidating on ECE). ECE before temperature scaling is indicated in brackets.

| Dataset | Model | CE | Brier Loss | MMCE | FLSD | DFL | AFL | GCE |
|---|---|---|---|---|---|---|---|---|
| CIFAR-10 | ResNet-50 | **0.66(4.25)** | 0.70(2.00) | 0.78(4.61) | 0.67(1.53) | 0.68(1.06) | 0.67(0.83) | **0.66(3.25)** |
| | ResNet-110 | 0.83(4.71) | 0.71(2.24) | 0.73(4.89) | **0.66(1.64)** | 0.75(1.57) | 0.77(0.84) | **0.66(3.69)** |
| | Wide-ResNet | **0.55(3.44)** | 0.68(1.23) | 0.68(3.60) | 0.59(1.98) | 0.67(1.01) | 0.56(0.61) | 0.68(2.66) |
| | DenseNet-121 | 0.81(4.63) | 0.83(1.72) | 0.79(4.98) | 0.75(1.42) | **0.70(0.95)** | 0.73(0.10) | 0.71(3.18) |
| CIFAR-100 | ResNet-50 | 1.75(17.49) | 1.52(4.73) | 1.53(15.52) | 1.69(5.40) | 1.53(2.06) | **1.39(1.71)** | 1.50(11.23) |
| | ResNet-110 | 2.06(18.78) | 1.65(6.04) | 2.11(18.62) | 1.60(6.94) | 1.49(4.28) | **1.29(1.46)** | 1.54(12.69) |
| | Wide-ResNet | 1.56(15.12) | 1.32(3.97) | 1.73(14.00) | 1.33(2.47) | 1.26(3.45) | 1.26(2.29) | **1.25(6.83)** |
| | DenseNet-121 | 2.09(19.27) | 1.58(4.21) | 1.94(17.08) | 1.55(3.31) | 1.56(4.79) | 1.51(1.71) | **1.48(10.75)** |
| Tiny-ImageNet | ResNet-50 | 1.59(16.52) | **0.82(3.89)** | 1.66(14.88) | 0.94(1.82) | 1.02(2.66) | 0.83(2.69) | 1.19(4.29) |
| | ResNet-110 | 1.62(16.07) | 1.50(4.15) | 1.61(16.25) | 1.03(2.22) | 1.04(2.55) | **0.97(2.74)** | 1.15(4.20) |
| AV1451 | MLP | 15.59(13.57) | 16.24(15.61) | 15.60(15.27) | 16.02(14.08) | **15.31(14.00)** | 17.64(15.85) | 15.62(13.01) |

contains a high proportion of ambiguous or difficult instances, where focal-style reweighting is particularly effective. In contrast, GCE regularizes predictions at the class-aggregated level, which mitigates overconfidence but does not specifically target the minority of tough examples. Nevertheless, it is important to note that GCE still substantially improves calibration compared to standard cross-entropy on Tiny-ImageNet. This highlights that the proposed loss effectively addresses the overconfidence issue even in highly challenging settings.

**Long-tailed robustness** Because GCE includes a class-level regularizer, its behavior under class imbalance warrants explicit examination. We therefore evaluate on CIFAR-10-LT (Cui et al., 2019), created by exponentially sub-sampling CIFAR-10 so that head classes contain many more samples than tail classes (imbalance factor $\rho$). Following standard practice, we consider multiple $\rho$ (e.g., 10–100) and report overall and classwise metrics (including confusion matrices). Full setup are provided in Appendix C. As reported in Table 3 and Table 4, as $\rho$ increases, all methods exhibit the expected accuracy drop due to reduced tail data; however, GCE shows less degradation beyond this common effect and its performance advantage over other losses becomes even more significant for larger $\rho$.

Table 3: Average test error (%) on CIFAR-10-LT computed for different losses.

| $\rho$ | Model | CE | Brier Loss | MMCE | FLSD | DFL | AFL | GCE |
|---|---|---|---|---|---|---|---|---|
| 10 | ResNet-50 | 10.55±0.32 | 10.90±0.44 | 10.55±0.37 | 11.44±0.31 | 11.04±0.33 | 11.71±0.29 | **10.28±0.18** |
| | ResNet-110 | 10.54±0.33 | 10.86±0.17 | 10.61±0.34 | 11.38±0.33 | 10.93±0.22 | 11.72±0.48 | **10.44±0.28** |
| | Wide-ResNet | 9.08±0.25 | 9.65±0.31 | 9.06±0.29 | 9.47±0.19 | 9.15±0.20 | 10.70±0.50 | **9.03±0.21** |
| | DenseNet121 | 10.23±0.31 | 10.03±0.28 | 10.11±0.26 | 11.16±0.59 | 10.46±0.30 | 11.52±0.30 | **10.06±0.25** |
| 100 | ResNet-50 | 26.62±1.02 | 31.05±1.29 | 26.69±0.95 | 28.43±1.14 | 26.89±1.29 | 29.92±1.07 | **25.35±1.09** |
| | ResNet-110 | 26.85±0.68 | 30.82±1.41 | 26.77±0.82 | 28.16±1.01 | 27.00±0.73 | 29.24±0.73 | **25.62±0.64** |
| | Wide-ResNet | 25.20±0.68 | 27.58±1.27 | 25.01±1.06 | 26.88±0.67 | 25.63±0.98 | 28.79±0.55 | **23.75±0.71** |
| | DenseNet121 | 26.02±0.87 | 25.30±0.91 | 26.07±1.19 | 26.84±1.19 | 26.21±1.23 | 28.92±1.90 | **23.79±0.63** |

Table 4: Average ClasswiseECE (%) on CIFAR-10-LT computed for different losses with adaptive temperature scaling (crossvalidating on ECE). ClasswiseECE before temperature scaling is indicated in brackets.

| $\rho$ | Model | CE | Brier Loss | MMCE | FLSD | DFL | AFL | GCE |
|---|---|---|---|---|---|---|---|---|
| 10 | ResNet-50 | 1.24 (1.87) | 1.24 (1.43) | 1.28 (1.86) | 1.34 (1.49) | 1.31 (1.58) | 1.36 (1.38) | **1.20 (1.83)** |
| | ResNet-110 | 1.23 (1.92) | 1.24 (1.48) | 1.22 (1.91) | 1.39 (1.53) | 1.28 (1.61) | 1.42 (1.44) | **1.22 (1.89)** |
| | Wide-ResNet | 1.08 (1.65) | 1.12 (1.22) | 1.08 (1.64) | 1.16 (1.18) | 1.14 (1.26) | 1.39 (1.40) | **1.07 (1.61)** |
| | DenseNet121 | 1.21 (1.84) | 1.12 (1.25) | 1.23 (1.79) | 1.30 (1.37) | 1.21 (1.43) | 1.31 (1.34) | **1.15 (1.80)** |
| 100 | ResNet-50 | 3.76 (5.00) | 3.95 (4.69) | 3.74 (4.99) | 4.15 (4.73) | 3.88 (4.61) | 4.54 (4.58) | **3.46 (4.73)** |
| | ResNet-110 | 3.75 (5.10) | 3.89 (4.73) | 3.75 (5.06) | 4.20 (4.74) | 3.86 (4.68) | 4.51 (4.54) | **3.45 (4.83)** |
| | Wide-ResNet | 3.71 (4.67) | 3.75 (4.26) | 3.64 (4.67) | 4.18 (4.40) | 3.90 (4.30) | 4.52 (4.55) | **3.37 (4.35)** |
| | DenseNet121 | 3.59 (4.83) | 3.41 (3.93) | 3.59 (4.85) | 4.01 (4.32) | 3.76 (4.38) | 4.31 (4.37) | **3.19 (4.36)** |

**Out-of-Distribution (OOD) detection** To test the robustness of GCE, following (Mukhoti et al., 2020), we evaluate GCE and other baseline methods on out-of-distribution (OOD) detection. Models are trained on CIFAR-10 as the in-distribution data and tested on SVHN (Netzer et al., 2011) and CIFAR-10-C (Hendrycks & Dietterich, 2019) with level-5 Gaussian noise corruption as OOD

Table 5: AUROC (%) of models trained on CIFAR-10 as the in-distribution data and tested on SVHN and CIFAR-10-C as out-of-distribution data.

| Dataset | Model | CE | Brier Loss | MMCE | FLSD | DFL | AFL | GCE |
|---------|-------|-----|-----------|------|------|-----|-----|-----|
| SVHN | ResNet-50 | 84.74 | 90.53 | 86.15 | 92.67 | 92.97 | 95.97 | 90.34 |
| | ResNet-110 | 81.33 | 92.17 | 76.49 | 90.72 | 90.85 | 95.91 | 89.58 |
| | Wide-ResNet | 92.61 | 94.83 | 91.09 | 91.44 | 88.70 | 96.93 | 93.05 |
| | DenseNet-121 | 88.43 | 89.64 | 80.58 | 92.04 | 90.79 | 95.44 | 92.44 |
| CIFAR-10-C | ResNet-50 | 87.26 | 88.19 | 86.87 | 88.64 | 86.51 | 92.40 | 90.52 |
| | ResNet-110 | 76.66 | 85.58 | 72.96 | 86.54 | 82.76 | 90.10 | 85.93 |
| | Wide-ResNet | 84.59 | 87.31 | 80.18 | 83.13 | 80.54 | 86.65 | 86.49 |
| | DenseNet-121 | 84.53 | 82.05 | 76.41 | 85.76 | 86.94 | 88.93 | 87.74 |

datasets. Using the entropy of the softmax output as the uncertainty score, we report the area under the receiver operating characteristic curve (AUROC) for OOD detection in Table 5. Models trained with GCE achieve substantial improvements over those trained with standard cross-entropy (CE), although they still fall behind AFL. This gap can be attributed to the fact that, while GCE combined with a post-hoc calibration method (e.g., temperature scaling on a held-out CIFAR-10 validation set) can reach in-distribution calibration performance comparable to AFL, post-hoc calibration degrades under distributional shift: the mapping learned from in-distribution validation data does not generalize reliably to shifted inputs, making entropy-based uncertainty estimates less effective.

**Computational Complexity**  From a complexity perspective, GCE augments the standard cross-entropy with a class-level normalization factor that is computed once per mini-batch and cached; this extra step is $\mathcal{O}(K)$ in both time and memory and can be fused into the existing softmax-loss kernel. Consequently, the forward–backward pass retains the same asymptotic cost as CE, and its runtime overhead is statistically negligible as shown in Table 6. In contrast, AFL incurs extra training time due to repeated validation-set binning and outer-loop updates of $\gamma$.

Table 6: Training-time for GCE and baselines across ResNet-50 & Wide-ResNet (CIFAR-10).

| Model | GCE | CE | Brier Loss | MMCE | FLSD | DFL | AFL |
|-------|-----|-----|-----------|------|------|-----|-----|
| ResNet50 | 16.65±0.33 | 16.47±0.34 | 16.22±0.45 | 17.08±0.28 | 16.88±0.29 | 16.51±0.41 | 18.77±0.38 |
| Wide-ResNet | 26.35±0.41 | 26.39±0.41 | 26.81±0.33 | 26.28±0.35 | 26.63±0.19 | 26.08±0.40 | 28.12±0.27 |

## 6 FUTURE DIRECTION

To explore whether the complementary strengths of GCE and focal loss can be captured simultaneously, we experimented with a simple hybrid by applying focal modulation to the GCE objective:

$$\mathcal{L}_{\text{GCE}} = -(1 - p_{y_i}^{(i)})^\gamma \frac{1}{N} \sum_{i=1}^{N} \log\left(\frac{p_{y_i}^{(i)}}{\sum_{j=1}^{N} \hat{p}_{y_i}^{(j)}}\right). \tag{6}$$

However, our experiments (see Appendix C) show that this straightforward combination performs poorly. Designing a more principled integration that can simultaneously improve accuracy, calibration, and robustness remains an open direction for future work.

## 7 CONCLUSION

In conclusion, GCE offers a simple yet effective approach to improving both the classification accuracy and calibration performance of deep neural networks. By introducing a class confidence regularization term, our method alleviates the overconfidence issue inherent in cross-entropy while preserving its strict properness. Moreover, we demonstrate that, when combined with appropriately designed post-hoc calibration techniques such as adaptive temperature scaling, GCE can achieve state-of-the-art calibration performance without compromising accuracy. Our contributions are supported by both rigorous theoretical analysis and extensive empirical validation, underscoring the superiority and broad applicability of GCE.

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

# A  PROOF OF THEOREM 1

*Proof.* We first define the conditional risk $\mathcal{R}_{\text{GCE}}$ for GCE as

$$\mathcal{R}_{\text{GCE}} = -\sum_{i=1}^{N}\sum_{k=1}^{K} q_k^{(i)} \log(\frac{\hat{p}_k^{(i)}}{\sum_{j=1}^{N}\hat{p}_k^{(j)}}). \tag{7}$$

Denote $n_k = \sum_{i=1}^{N} q_k^{(i)}$ and $s_k = \sum_{i=1}^{N} \hat{p}_k^{(i)}$, $\mathcal{R}_{\text{GCE}}$ can be rewritten as

$$
\begin{aligned}
\mathcal{R}_{\text{GCE}} &= -\sum_{k=1}^{K}\sum_{i=1}^{N} q_k^{(i)} \log(\frac{\hat{p}_k^{(i)}}{\sum_{j=1}^{N}\hat{p}_k^{(j)}}) \\
&= -\sum_{k=1}^{K} n_k \sum_{i=1}^{N} \frac{q_k^{(i)}}{n_k} \log(\frac{\hat{p}_k^{(i)}}{s_k}) \\
&= -\sum_{k=1}^{K} n_k H(Q_k, P_k),
\end{aligned}
\tag{8}
$$

where $Q_k = (\frac{q_k^{(1)}}{n_k}, \cdots, \frac{q_k^{(N)}}{n_k})$, $P_k = (\frac{\hat{p}_k^{(1)}}{s_k}, \cdots, \frac{\hat{p}_k^{(N)}}{s_k})$ and $H(Q, P)$ is the cross-entropy between two distribution $Q$ and $P$. Recall that

$$H(Q, P) = H(Q, Q) + D_{\text{KL}}(Q||P), \tag{9}$$

by the non-negativity of the Kullback–Leibler divergence $D_{\text{KL}}(Q||P)$, we know that $H(Q, P) \geq H(Q, Q)$ with equality holds if and only if $Q = P$. Insert this equality into Eq. equation 8, we have

$$\mathcal{R}_{\text{GCE}} \geq -\sum_{k=1}^{K} n_k H(Q_k, Q_k), \tag{10}$$

with equality holds if and only if $Q_k = P_k$, $\forall k$, i.e.,

$$\hat{p}_k^{(i)} = s_k \frac{q_k^{(i)}}{n_k}, \ \forall i, k. \tag{11}$$

Substituting Eq. equation 11 into the constraint $\sum_{k=1}^{k} \hat{p}_k^{(i)} = 1$ yields $\sum_{k=1}^{K} s_k \frac{q_k^{(i)}}{n_k} = 1, \forall i \in [N]$.

This is a linear system in $\mathbf{s} = (s_1, \cdots, s_k)^\top$ and can be reformulated in matrix form

$$\mathbf{A}\mathbf{s} = \mathbf{1}_N, \tag{12}$$

where $\mathbf{A} = \mathbf{Q}^\top \text{diag}(n_1, \cdots, n_K)$. Since $\mathbf{Q}$ has full row rank, so does $\mathbf{A}$, so Eq. equation 12 has at most one solution. Clearly $s_k = n_k$, $k = 1, \cdots, K$ solves Eq. equation 11, thus unique. Substituting back gives $\hat{p}_k^{(i)} = q_k^{(i)}$, $\forall i, k$, implying the generative cross-entropy is strictly proper. □

# B  DATASET DESCRIPTION AND IMPLEMENTATION DETAILS

We use the following datasets in our experiments:

**CIFAR-10/100**  CIFAR-10 comprises 60,000 RGB images of size 32±32, evenly split across 10 categories (6,000 per class), with 50,000 training and 10,000 test images. CIFAR-100 follows the same format but extends to 100 categories (600 images per class). Following common practice, we hold out 5,000 images from the CIFAR-100 training set for validation.

**Tiny-ImageNet**  This dataset is a downscaled subset of ImageNet used in the ILSVRC challenge. It contains 100,000 color images from 200 classes (500 images per class), all resized to 64±64 pixels.

**SVHN** The Street View House Numbers (SVHN) dataset captures real-world digit recognition from Google Street View. It provides over 600,000 images labeled into 10-digit classes. We report results on the official test split to assess performance under dataset shift.

**CIFAR-10-C** CIFAR-10-C augments CIFAR-10 with diverse common corruptions applied at five severity levels. In our setup, the first 10,000 images correspond to the test set corrupted at severity level 1, and the last 10,000 to severity level 5. Unless otherwise noted, we evaluate robustness using the Gaussian Noise corruption at severity 5.

**AV1451** The AV-1451 ($^{18}$F-Flortaucipir) dataset comprises 1,080 pre-processed tau-PET scans, each paired with a T1 MRI and annotated with a three-class diagnosis label—cognitively normal, mild cognitive impairment, or Alzheimer's disease. After standard motion correction, co-registration, and SUVR normalisation, every scan is distilled into a fixed 68-dimensional feature vector capturing regional tau uptake across FreeSurfer-defined cortical and subcortical ROIs. For modelling we adopt an $0.8/0.1/0.1$ split: 864 samples for training, 108 for validation, and 108 held out for final testing, ensuring subject-level independence across splits.

## C Extra Experiment Results

**Extra Calibration Metrics** Here we provide the average AdaECE (Table 7) and average Classwise-ECE (Table 8) over 10 runs. All metrics are computed using 20 bins.

Table 7: Average AdaECE (%) over 10 runs computed for different losses with adaptive temperature scaling (crossvalidating on ECE). AdaECE before temperature scaling is indicated in brackets.

| Dataset | Model | CE | Brier Loss | MMCE | FLSD | DFL | AFL | GCE |
|---------|-------|-----|-----------|------|------|-----|-----|-----|
| CIFAR-10 | ResNet-50 | 0.83(4.23) | 0.65(2.04) | 1.14(4.60) | 0.64(1.75) | 0.71(1.22) | 0.58(0.69) | 0.65(3.21) |
| | ResNet-110 | 1.15(4.69) | 0.69(2.21) | 1.26(4.88) | 0.65(1.78) | 0.74(1.64) | 0.62(0.71) | 0.82(3.68) |
| | Wide-ResNet | 0.72(3.41) | 0.66(1.77) | 0.78(3.57) | 0.70(1.89) | 0.71(1.49) | 0.51(0.51) | 0.70(2.62) |
| | DenseNet-121 | 1.04(4.62) | 0.86(1.99) | 1.17(4.97) | 0.82(1.38) | 0.77(0.99) | 0.59(0.86) | 0.73(3.17) |
| CIFAR-100 | ResNet-50 | 1.85(17.48) | 1.17(4.59) | 1.51(15.49) | 1.37(5.31) | 1.26(1.81) | 1.05(1.53) | 1.30(11.18) |
| | ResNet-110 | 1.86(18.77) | 1.34(5.88) | 2.02(18.61) | 1.31(6.90) | 1.17(4.46) | 1.09(1.26) | 1.29(12.68) |
| | Wide-ResNet | 1.91(15.09) | 1.43(3.89) | 1.99(13.95) | 1.25(2.41) | 1.35(3.37) | 1.32(2.35) | 1.21(6.71) |
| | DenseNet-121 | 2.45(19.26) | 1.53(4.07) | 2.15(17.07) | 1.58(3.18) | 1.56(4.85) | 1.50(1.73) | 1.68(10.72) |
| Tiny-ImageNet | ResNet-50 | 1.57(16.50) | 1.04(3.09) | 1.68(14.86) | 0.92(1.79) | 1.13(2.59) | 1.00(2.67) | 1.19(4.32) |
| | ResNet-110 | 1.69(16.05) | 1.69(2.27) | 1.62(16.24) | 0.94(2.14) | 1.06(2.49) | 1.04(2.70) | 1.28(4.22) |
| AV1451 | MLP | 22.63(20.97) | 22.11(20.70) | 21.38(20.84) | 22.27(22.59) | 20.38(19.36) | 22.03(21.16) | 22.30(20.30) |

Table 8: Average ClasswiseECE (%) over 10 runs computed for different losses with adaptive temperature scaling (crossvalidating on ECE). ECE before temperature scaling is indicated in brackets.

| Dataset | Model | CE | Brier Loss | MMCE | FLSD | DFL | AFL | GCE |
|---------|-------|-----|-----------|------|------|-----|-----|-----|
| CIFAR-10 | ResNet-50 | 0.50(0.89) | 0.44(0.49) | 0.56(0.96) | 0.43(0.46) | 0.44(0.45) | 0.35(0.35) | 0.42(0.71) |
| | ResNet-110 | 0.56(0.98) | 0.46(0.54) | 0.59(1.01) | 0.43(0.46) | 0.46(0.47) | 0.36(0.35) | 0.47(0.79) |
| | Wide-ResNet | 0.43(0.73) | 0.43(0.43) | 0.45(0.76) | 0.36(0.49) | 0.38(0.39) | 0.33(0.32) | 0.39(0.59) |
| | DenseNet-121 | 0.50(0.96) | 0.46(0.49) | 0.57(1.03) | 0.41(0.47) | 0.44(0.46) | 0.36(0.37) | 0.44(0.69) |
| CIFAR-100 | ResNet-50 | 0.23(0.39) | 0.23(0.22) | 0.23(0.26) | 0.22(0.22) | 0.22(0.23) | 0.21(0.22) | 0.22(0.28) |
| | ResNet-110 | 0.23(0.41) | 0.23(0.24) | 0.23(0.41) | 0.22(0.24) | 0.23(0.23) | 0.20(0.21) | 0.22(0.31) |
| | Wide-ResNet | 0.22(0.34) | 0.21(0.21) | 0.22(0.32) | 0.20(0.20) | 0.20(0.20) | 0.20(0.22) | 0.20(0.21) |
| | DenseNet-121 | 0.24(0.42) | 0.22(0.22) | 0.24(0.39) | 0.22(0.22) | 0.22(0.26) | 0.21(0.22) | 0.22(0.28) |
| Tiny-ImageNet | ResNet-50 | 0.19(0.25) | 0.18(0.19) | 0.19(0.24) | 0.18(0.18) | 0.18(0.18) | 0.18(0.18) | 0.18(0.19) |
| | ResNet-110 | 0.18(0.24) | 0.19(0.19) | 0.18(0.24) | 0.18(0.18) | 0.18(0.18) | 0.18(0.18) | 0.19(0.19) |
| AV1451 | MLP | 14.57(14.73) | 13.44(12.79) | 14.05(14.62) | 15.96(12.86) | 14.59(14.02) | 13.90(12.98) | 14.01(14.20) |

**Bar Plots** Extra bar plots of test error and ECE are presented in Figure 3.

**One-sided Wilcoxon Signed Rank Test** Here we report p-values of one-sided Wilcoxon signed rank test, which show that GCE significantly improves classification accuracy on CIFAR-10/100 compared to other baselines.

**Long-Tailed Robustness** To evaluate GCE under class imbalance, we constructed CIFAR-10-LT, a long-tailed version of the original 50,000-image CIFAR-10 training split. Given an imbalance

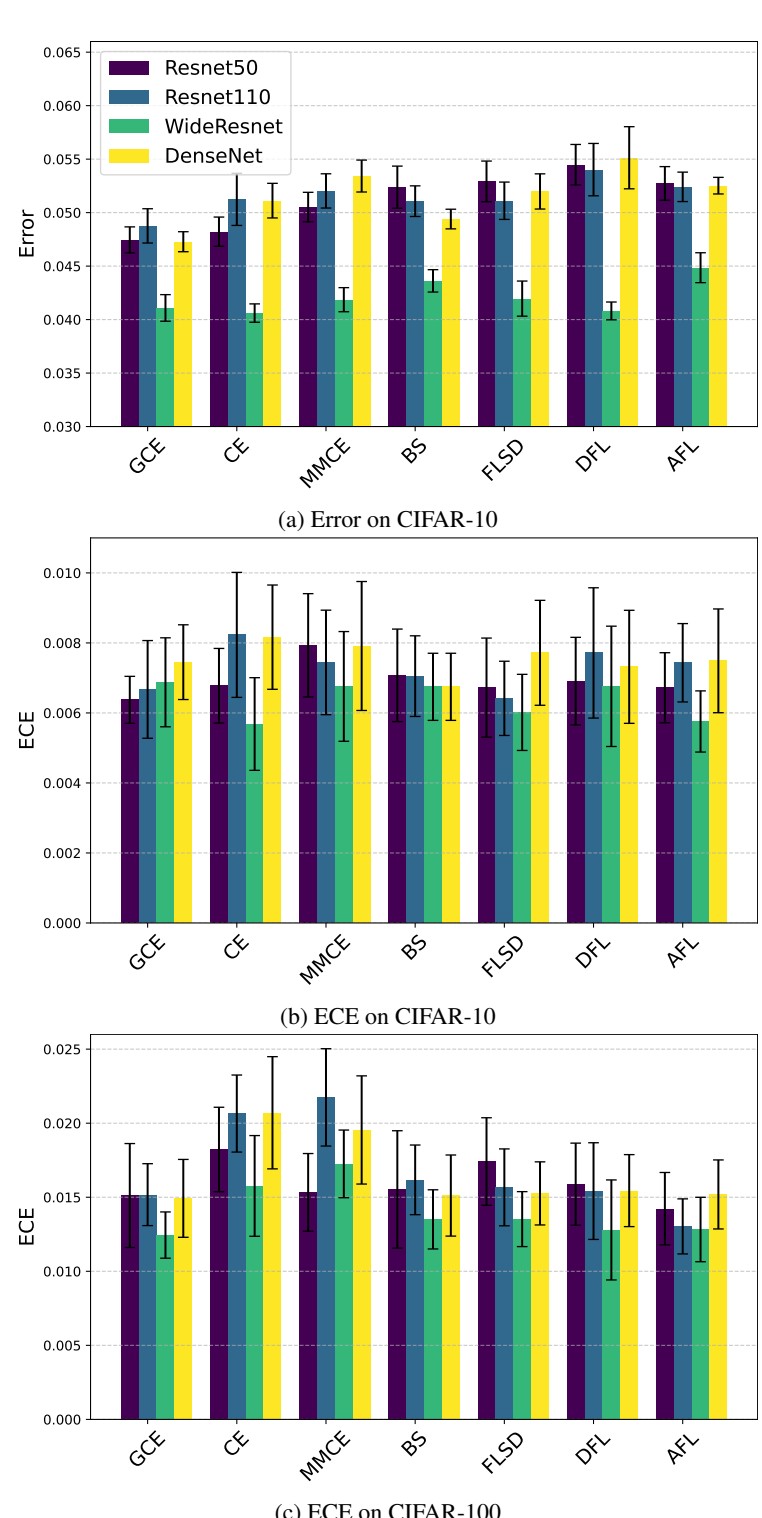

(a) Error on CIFAR-10

(b) ECE on CIFAR-10

(c) ECE on CIFAR-100

Figure 3: Classification error and ECE with 95% confidence intervals across four architectures on CIFAR-10/100.

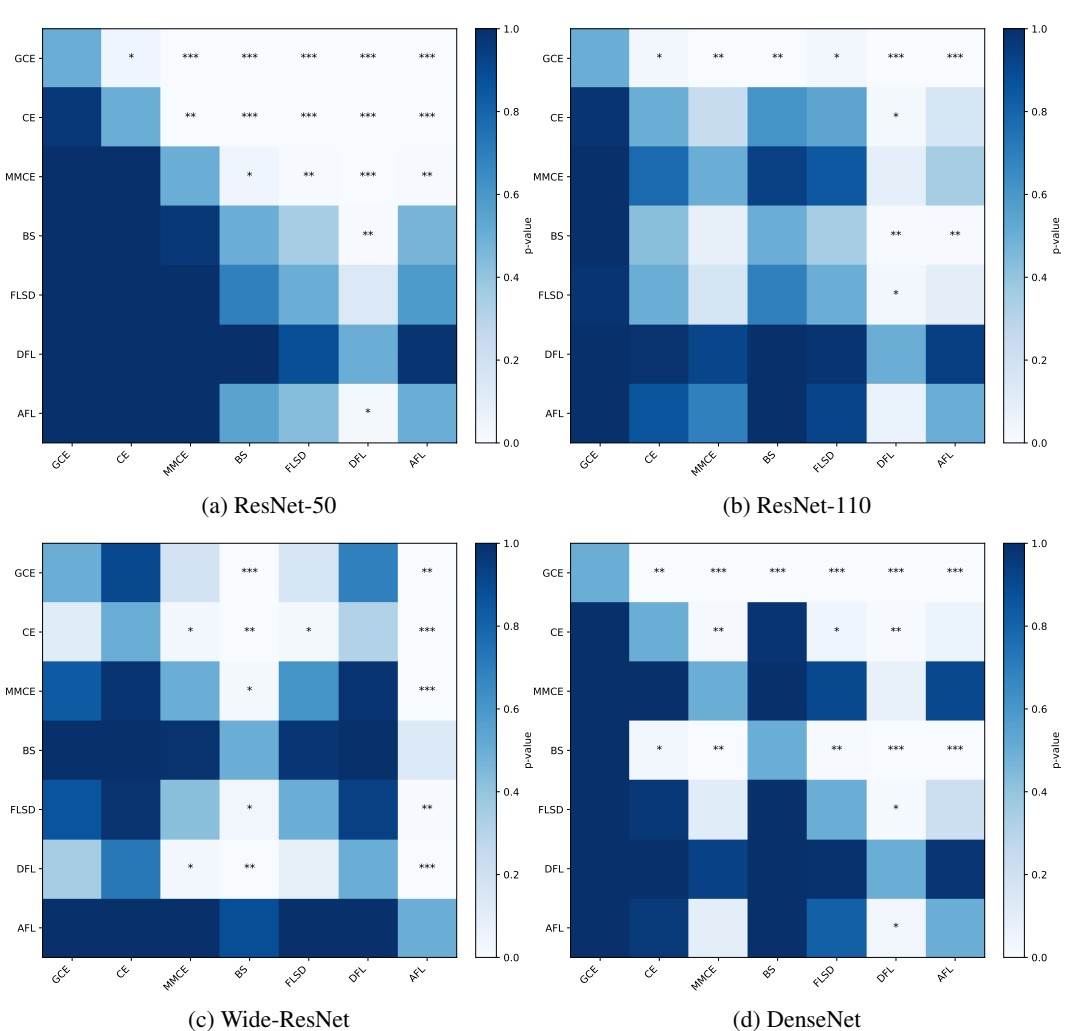

(a) ResNet-50

(b) ResNet-110

(c) Wide-ResNet

(d) DenseNet

Figure 4: One-sided Wilcoxon p-value heatmap on CIFAR-10 for test error. $H_1$: row $<$ column; each cell uses $n = 10$ paired runs. Asterisks denote significance ($* : p < 0.05, ** : p < 0.01, *** : p < 0.001$).

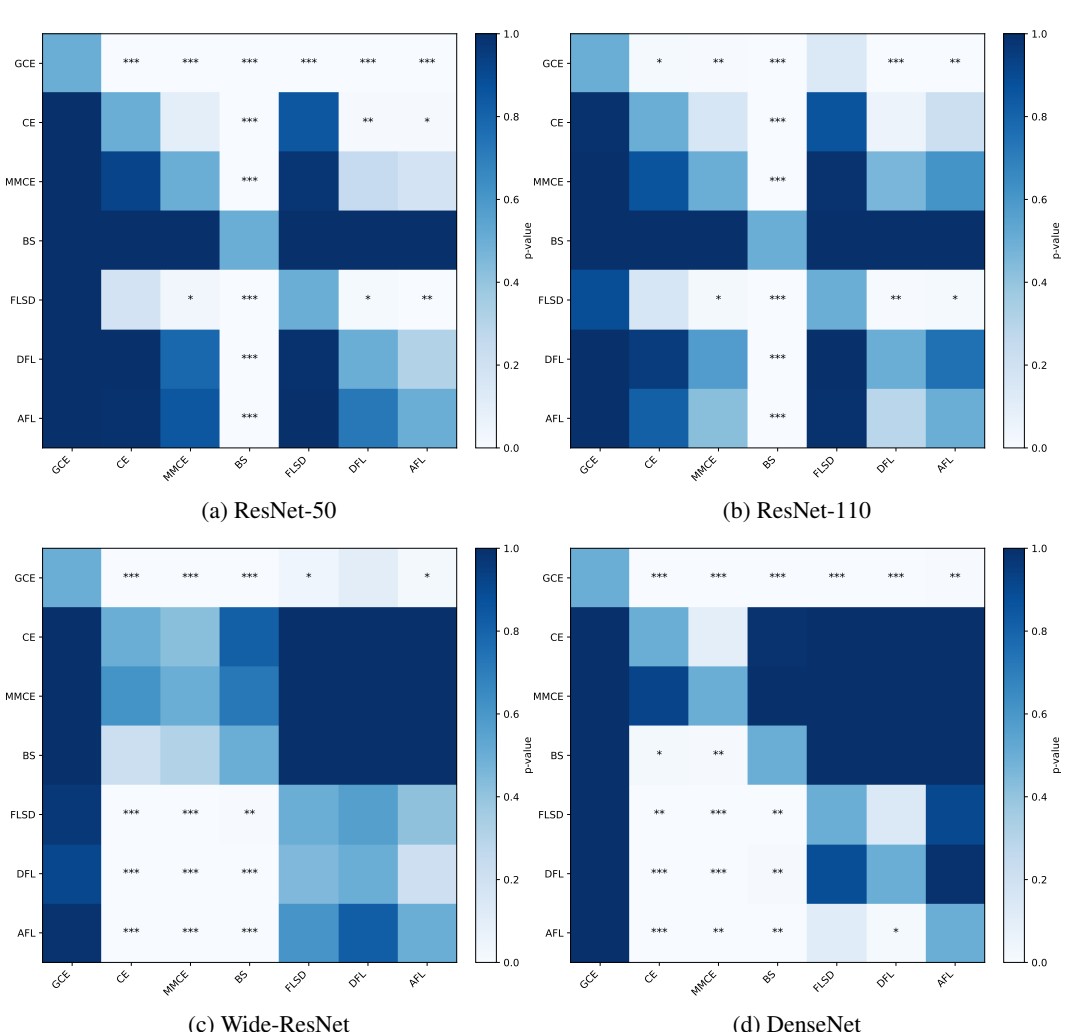

Figure 5: One-sided Wilcoxon p-value heatmap on CIFAR-100 for test error. $H_1$: row < column; each cell uses $n = 10$ paired runs. Asterisks denote significance ($* : p < 0.05, ** : p < 0.01, *** : p < 0.001$).

factor $\rho$—the ratio of the largest to the smallest class—we (i) designate one class as the head and retain all of its 5 000 images; (ii) rank the remaining nine classes from less to more scarce ($k = 1, \ldots, 9$) and allocate to class $k$ with $n_k = 5000\rho^{-k/9}$, rounding to the nearest integer; (iii) sample those $n_k$ images uniformly without replacement under a fixed random seed for reproducibility. The 10,000-image test set is left unchanged, preserving a balanced evaluation protocol. By setting $\rho \in \{10, 100\}$ we create gaps of one and two orders of magnitude, respectively, between the head and tail classes, thereby turning CIFAR-10 into a controllable long-tailed benchmark. Under both imbalance settings ($\rho = 10$ and 100), GCE consistently outperforms all baseline losses, demonstrating superior robustness in the presence of severe head-tail disparity.

**Combination of GCE and Focal Loss**   We attempt to combine GCE with adaptive focal loss to capture the complementary strengths of both methods. We experimented with a simple hybrid by applying focal modulation to the GCE objective:

$$\mathcal{L}_{\text{GCE}} = -(1 - p_{y_i}^{(i)})^\gamma \frac{1}{N} \sum_{i=1}^{N} \log\Big(\frac{p_{y_i}^{(i)}}{\sum_{j=1}^{N} \hat{p}_{y_i}^{(j)}}\Big). \tag{13}$$

where $\gamma$ is updated according to the same rule of AFL, the result (Table 9) shows that this combination does not improve the performance compared to the original AFL.

Table 9: Performance of the combination of GCE and focal loss over five runs with adaptive temperature scaling (cross-validating on ECE).

| Model | Method | Test error | ECE | AdaECE | ClasswiseECE |
|---|---|---|---|---|---|
| ResNet-50 | AFL | 48.61±0.28 | 0.83±0.17 | 1.04±0.32 | 0.18±0.00 |
| | GCE | 50.86±0.33 | 1.34±0.14 | 1.71±0.21 | 0.19±0.00 |
| | AFL+GCE | 50.79±0.57 | 0.84±0.22 | 1.17±0.18 | 0.18±0.00 |
| ResNet-110 | AFL | 46.68±0.26 | 1.25±0.30 | 1.14±0.15 | 0.18±0.00 |
| | GCE | 49.47±0.32 | 1.53±0.27 | 1.57±0.22 | 0.19±0.00 |
| | AFL+GCE | 49.60±0.70 | 0.72±0.11 | 1.01±0.27 | 0.18±0.00 |

**Number of bins used for Adaptive Temperature Scaling**   We compare the performance of adaptive temperature scaling trained with 5, 10, 15, 20, and 25 equal mass bins; the result is reported in Table 10. We observe50 that the best results are for the number of bins $> 10$. We choose bin $= 15$ to balance the efficiency and the performance.

Table 10: ECE (%) performance for ResNet-50 trained on CIFAR-10 when Adaptive Temperature scaling training uses a different number of equal-mass bins.

| Metric | Bins | CE | Brier Loss | MMCE | FLSD | DFL | AFL | GCE |
|---|---|---|---|---|---|---|---|---|
| ECE | 5 | 1.28 | 0.74 | 1.45 | 0.73 | 0.62 | 0.67 | 0.74 |
| | 10 | 0.72 | 0.76 | 0.82 | 0.67 | 0.64 | 0.67 | 0.67 |
| | 15 | 0.66 | 0.70 | 0.78 | 0.67 | 0.68 | 0.67 | 0.66 |
| | 20 | 0.64 | 0.73 | 0.83 | 0.69 | 0.68 | 0.64 | 0.65 |
| | 25 | 0.60 | 0.74 | 0.73 | 0.65 | 0.61 | 0.67 | 0.66 |
| AdaECE | 5 | 1.17 | 0.66 | 1.61 | 0.75 | 0.68 | 0.54 | 0.69 |
| | 10 | 0.79 | 0.66 | 1.03 | 0.66 | 0.72 | 0.58 | 0.63 |
| | 15 | 0.83 | 0.65 | 1.14 | 0.64 | 0.71 | 0.58 | 0.65 |
| | 20 | 0.81 | 0.65 | 1.04 | 0.68 | 0.67 | 0.56 | 0.70 |
| | 25 | 0.76 | 0.68 | 1.03 | 0.68 | 0.69 | 0.57 | 0.70 |
| ClasswiseECE | 5 | 0.50 | 0.44 | 0.56 | 0.44 | 0.45 | 0.35 | 0.42 |
| | 10 | 0.49 | 0.44 | 0.56 | 0.43 | 0.44 | 0.35 | 0.42 |
| | 15 | 0.50 | 0.44 | 0.56 | 0.43 | 0.44 | 0.35 | 0.42 |
| | 20 | 0.50 | 0.44 | 0.56 | 0.44 | 0.44 | 0.35 | 0.41 |
| | 25 | 0.50 | 0.44 | 0.56 | 0.43 | 0.44 | 0.35 | 0.41 |

## D    USAGE OF LARGE LANGUAGE MODELS (LLMS)

We used large language models as a general-purpose writing assistant. Its role was limited to grammar checking, minor stylistic polishing, and improving the clarity of phrasing in some parts of the manuscript. The authors made all substantive contributions to the research and writing.

