# OpenReview forum: "Towards Accurate and Calibrated Classification: Regularizing Cross-Entropy From A Generative Perspective"
_ICLR.cc/2026/Conference — ICLR 2026 Conference Withdrawn Submission_

### Official Review · Reviewer_wRsu · 2025-10-22

**Soundness:** 4
**Presentation:** 4
**Contribution:** 4
**Rating:** 6
**Confidence:** 5

**Summary:**

The paper introduces Generative Cross-Entropy (GCE), a novel loss function that approaches calibration from a generative perspective rather than the traditional discriminative perspective. GCE is shown to be equivalent to cross-entropy augmented with a class-level confidence regularizer. The method claims to improve both classification accuracy and probability calibration simultaneously, especially on long-tailed scenarios.

**Strengths:**

- The paper is well-written and easy to follow along.
- The generative perspective seem to provide some principled approach to calibration.
- Offers a proof for the strictly proper loss property of the GCE loss.
- Reasonable good performance overall and a good computational analaysis of the method.

**Weaknesses:**

- The paper does does not include a few important papers in literature in the body of research, a couple of examples are below
  - https://arxiv.org/abs/1912.02781
  - https://proceedings.neurips.cc/paper_files/paper/2019/hash/36ad8b5f42db492827016448975cc22d-Abstract.html
- GCE simply has an extra regularization term on top of the standard CE, then the hunch is that in its current form without adaptive temperature scaling it might have negligible impact on calibration and/or accuracy.
- There are other losses that guarantee regularization effect on top of the standard CE loss and thus the better calibration/accuracy tradeoffs. Those losses include mixup, label-smoothing (https://arxiv.org/abs/1906.02629), the paper does not make an effort to cross validate with such techniques either.
- Lastly, it is clearly understandable that there is a theoretical guarantee with P = Q in the proposed methodology, however, the current intuition indicates that the bin level confidences and probably to some extent the regularization term in GCE are contributing to the performance  benefits, therefore, the paper requires some clarifications on the real impacts of each of the components of the proposed method.

**Questions:**

- Especially for long-tail cases, we observe almost insignificant difference (Table 3) between GCE and CE. Therefore, it is apparent to understand the impact of Adaptive temperature scaling and GCE when operate independently. Following are a few perspectives that require answers
  - Conduct a sensitivity analysis on the absence of Adaptive temperature scaling in GCE?
  - What if AFL is combined with adaptive temperature scaling and/or GCE? Will the performance get better in any of these ablations? Will there be a surprise saying the so called generative perspective is more artificial and the regularization is the root cause behind the benefits?

- The advantages of being a *strictly proper loss* are unclear from the paper. What if it is not a strictly proper loss and still do the job?

---

### Official Review · Reviewer_T2wL · 2025-10-31

**Soundness:** 3
**Presentation:** 2
**Contribution:** 3
**Rating:** 4
**Confidence:** 4

**Summary:**

This paper introduces Generative Cross-Entropy (GCE), a strictly proper loss under mild conditions, designed to improve both classification accuracy and calibration of deep neural networks. GCE augments the standard cross-entropy (CE) loss with a class-level confidence regularizer derived from a generative formulation.
Additionally, the paper proposes an adaptive piecewise temperature scaling (ATS) method for post-hoc calibration.
Experiments across CIFAR-10/100, Tiny-ImageNet, and a medical imaging dataset show that GCE marginally improves both accuracy and calibration, with computational cost comparable to CE.

**Strengths:**

- The strict properness analysis provides theoretical justification missing from many ad-hoc calibration losses.
- The decomposition in Eq. 5 provides clear intuition about how GCE differs from standard CE.
- Comprehensive empirical evaluation across multiple datasets and architectures, with shown improvements in both accuracy and calibration. Statistical tests and multiple random seeds are used. Experiments on OOD detection are included.

**Weaknesses:**

**Missing related work**

The paper would benefit from a more complete discussion of recent literature on proper calibration errors and relationship between focal loss and properness (since the paper mentions that focal loss is not strictly proper in L156), e.g. papers like:

https://proceedings.mlr.press/v238/popordanoska24a/popordanoska24a.pdf

https://arxiv.org/pdf/2408.11598

**Overstated claims**

Conclusion states: “GCE can achieves state-of-the-art calibration performance”. This is only true after applying ATS. The raw GCE calibration is clearly worse than focal variants (Table 2, numbers in brackets – before TS).

**Overstated “generative” interpretation**

The use of the term “generative” is somewhat misleading, as the method does not explicitly model p(x|y); moreover, the writing at times overstates the novelty, with phrases such as “maximizing p(x|y)” that imply a deeper link to generative modeling, even though the proposed approach is essentially a cross-entropy loss augmented with a class-level confidence regularizer.

**Modest empirical gains**

Improvements over strong baselines (e.g., AFL) are relatively small and sometimes dataset-dependent. Furthermore, the intuition that the proposed loss “regularizes class-level confidence, thereby discouraging overconfident predictions” is reasonable, but the empirical evidence in Table 8 does not clearly support this claim. Across datasets and architectures, GCE’s classwise ECE improvements are marginal and often comparable to or weaker than those of other baselines such as AFL or DFL. If the second term is intended to meaningfully regulate class-level confidence, I would expect more consistent reductions in classwise calibration error.

**Limitations of the method not discussed**

Minor:
- I would suggest summarizing the main contributions clearly at the end of the introduction to improve clarity of the paper
- Introduction doesn’t clearly state that the main contribution is the Generative Cross-Entropy (GCE)
- L185: delta should be defined
- Eq. (4) Notation should be clarified in the text
- Details about what binning scheme (equal width or equal mass binning) missing in main paper (I would suggest mentioning that 20 bins were used in Evaluations paragraph, instead of Sec. 5.2).

**Questions:**

How sensitive is ATS to its hyperparameters?

---

### Official Review · Reviewer_JNqp · 2025-11-01

**Soundness:** 2
**Presentation:** 3
**Contribution:** 2
**Rating:** 2
**Confidence:** 4

**Summary:**

The paper tackles the limitations of focal-loss based calibration approaches where they potentially compromise the predictive performance of a model for the sake of improving model calibration. To this end, the paper introduces a generative perspective into the discriminative classification training. Specifically, a loss term is incorporated with the traditional cross-entropy (CE) loss which acts a class-level confidence regularizer and aims to avoid overfitting, thereby claiming to reduce overconfident behavior. Further, the paper proposes adaptive temperature scaling technique which learns temperature parameters according to the diversity of training samples. Experiments are performed on CIFAR-10 /100, Tiny-ImageNet dataset and a medical dataset. Results claim to show competitive performance to CE baseline and focal loss variants in predictive accuracy and calibration error.

**Strengths:**

- The problem of losing predictive performance upon aiming to improving model calibration in focal loss variants is a relevant problem.

- The paper introduces a regularizer term based on approximating the posterior probability along side the traditional CE loss, which is simple to implement.

- The idea of adaptive temperature scaling is gradient-free.

**Weaknesses:**

- The experimental results are lacking in many aspects:

   - more dataset comparisons are required such as ImageNet.

   - the evaluation comparison is missing recent methods such as [A,B,C,D,E], which undermines the comparative advantage of the    proposed approach.

    - the ECE (table 2) is quite poor in many instances for GCE without using the adaptive temperature scaling, which raises concerns on the effectiveness of GCE.

    - the paper is missing reliability diagrams which is an important visualization to show the overconfidence and underconfidence reducing capabilities of a method.

    -  the method provides mixed improvements and the results are not consistently better than competing methods.


- It is hard to establish a connection between the two contributions of a regularizer and an adaptive temperature scaling post-hoc idea. Also, since GCA is a train-time regularizer and ATS is a post-hoc method, and GCA could barely seems to be effective for reducing miscalibration, there seems to be clear gap in the overall set of technical contributions.

- It is not much clear how GCA is helping to reduce miscalibration. Specifically, analyses to establish the grounding of GCA is missing.

- Will GCA be effective with other post-hoc methods such as simple temparature scaling and other methods such as [F]

- L207: The paper claims that inclusion of second term reduces overfitting, however, there is no (empirical) evidence to this claim. A



[A] Park, H., Noh, J., Oh, Y., Baek, D. and Ham, B., 2023. Acls: Adaptive and conditional label smoothing for network calibration. In Proceedings of the IEEE/CVF International Conference on Computer Vision (pp. 3936-3945).

[B] Liu, J., Ye, C., Cui, R. and Barnes, N., 2024. Self-calibrating vicinal risk minimisation for model calibration. In Proceedings of the IEEE/CVF Conference on Computer Vision and Pattern Recognition (pp. 3335-3345).

[C] Ni, J., Zhao, H., Gao, J., Guo, D. and Zha, H., 2025. Balancing Two Classifiers via A Simplex ETF Structure for Model Calibration. In Proceedings of the Computer Vision and Pattern Recognition Conference (pp. 30712-30721).

[D] Liu, B., Ben Ayed, I., Galdran, A. and Dolz, J., 2022. The devil is in the margin: Margin-based label smoothing for network calibration. In Proceedings of the IEEE/CVF Conference on Computer Vision and Pattern Recognition (pp. 80-88).

[E] Hebbalaguppe, R., Prakash, J., Madan, N. and Arora, C., 2022. A stitch in time saves nine: A train-time regularizing loss for improved neural network calibration. In Proceedings of the IEEE/CVF Conference on Computer Vision and Pattern Recognition (pp. 16081-16090).

[F] Zadrozny, B. and Elkan, C., 2002, July. Transforming classifier scores into accurate multiclass probability estimates. In Proceedings of the eighth ACM SIGKDD international conference on Knowledge discovery and data mining (pp. 694-699).

**Questions:**

- Can the proposed GCE also be applied to vision transformer based networks will it be effective with and without ATS?

- L90-91: needs references

- L29-34: needs references

---

### Official Review · Reviewer_VHsP · 2025-11-02

**Soundness:** 3
**Presentation:** 3
**Contribution:** 2
**Rating:** 2
**Confidence:** 5

**Summary:**

The confidence calibration of deep neural networks is considered. The paper introduces Generative Cross-Entropy (GCE), a novel loss function for deep neural networks that addresses the persistent trade-off between classification accuracy and confidence calibration. GCE is derived from a generative perspective by maximizing the posterior likelihood p(x∣y), which is shown to be equivalent to cross-entropy augmented with a class-level confidence regularizer that mitigates overconfidence. The method is proven to be strictly proper and consistently decrease the calibration error across multiple datasets and architectures.

**Strengths:**

1. The work proposed the generative cross-entropy loss and established its strict properness.

**Weaknesses:**

1. The training-time regularization-based methods have recently been extensively researched in the confidence calibration community. Discussion and comparison of existing techniques, such as CRL [A] (ICML'20), CPC [B] (CVPR'22), CALS [C] (CVPR'23), ACLS [D] (ICCV'23) etc, are missing. Without comparison against these approaches, it is groundless to claim that "GCE can achieve state-of-the-art calibration performance" (L483).

[A] Confidence-aware learning for deep neural networks. ICML'20.
[B] Calibrating Deep Neural Networks by Pairwise Constraints. CVPR'22.
[C] Class Adaptive Network Calibration. CVPR'23.
[D] ACLS: Adaptive and Conditional Label Smoothing for Network Calibration. ICCV'23.

**Questions:**

Inappropriate cites.
-- L168: The fact that cross-entropy is strictly proper is established long before Charoenphakdee et al., 2021).
-- The origination of the concept of strict proper loss should be cited.

Typos
L641: "32 \pm 32" -> "32 \times 32"

---

### Note · Authors · 2026-01-18

I have read and agree with the venue's withdrawal policy on behalf of myself and my co-authors.